# Anti-Inflammatory Activity and Mechanism of Isookanin, Isolated by Bioassay-Guided Fractionation from *Bidens pilosa* L.

**DOI:** 10.3390/molecules26020255

**Published:** 2021-01-06

**Authors:** Ying-Ji Xin, Soojung Choi, Kyung-Baeg Roh, Eunae Cho, Hyanggi Ji, Jin Bae Weon, Deokhoon Park, Wan Kyunn Whang, Eunsun Jung

**Affiliations:** 1Biospectrum Life Science Institute, Yongin 16827, Korea; bioua@biospectrum.com (Y.-J.X.); biocm@biospectrum.com (S.C.); biosh@biospectrum.com (K.-B.R.); biozr@biospectrum.com (E.C.); biocr@biospectrum.com (H.J.); biohy@biospectrum.com (J.B.W.); pdh@biospectrum.com (D.P.); 2Department of Global Innovative Drug, Graduate School, College of Pharmacy, Chung-Ang University, Heukseok-dong, Dongjak-gu, Seoul 156-756, Korea

**Keywords:** isookanin, *Bidens pilosa* L., bioassay-guided isolation, anti-inflammatory

## Abstract

*Bidens pilosa* L. (Asteraceae) has been used historically in traditional Asian medicine and is known to have a variety of biological effects. However, the specific active compounds responsible for the individual pharmacological effects of *Bidens pilosa* L. (*B. pilosa*) extract have not yet been made clear. This study aimed to investigate the anti-inflammatory phytochemicals obtained from *B. pilosa*. We isolated a flavonoids-type phytochemical, isookanin, from *B. pilosa* through bioassay-guided fractionation based on its capacity to inhibit inflammation. Some of isookanin’s biological properties have been reported; however, the anti-inflammatory mechanism of isookanin has not yet been studied. In the present study, we evaluated the anti-inflammatory activities of isookanin using lipopolysaccharide (LPS)-stimulated RAW 264.7 macrophages. We have shown that isookanin reduces the production of proinflammatory mediators (nitric oxide, prostaglandin E_2_) by inhibiting the expression of inducible nitric oxide synthase (iNOS) and cyclooxygenase-2 (COX-2) in LPS-stimulated macrophages. Isookanin also inhibited the expression of activator protein 1 (AP-1) and downregulated the LPS-induced phosphorylation of p38 mitogen-activated protein kinase (MAPK) and c-jun NH_2_-terminal kinase (JNK) in the MAPK signaling pathway. Additionally, isookanin inhibited proinflammatory cytokines (tumor necrosis factor-a (TNF-α), interleukin-6 (IL-6), interleukin-8 (IL-8), and interleukin-1β (IL-1β)) in LPS-induced THP-1 cells. These results demonstrate that isookanin could be a potential therapeutic candidate for inflammatory disease.

## 1. Introduction

Inflammation is a key function of the immune system in response to an injury, irritation, or noxious chemicals and it is necessary to maintain homeostasis in normal tissues. However, excessive inflammation is intimately related to the pathogenesis of various diseases, including arthritis, atherosclerosis, asthma, obesity, cancer, and autoimmune diseases [1]. Lipopolysaccharide (LPS), a component of the outer membrane of gram-negative bacteria, is recognized by toll-like receptors (TLRs) and its inducement is considered to be a key process leading to an inflammatory response [2]. It is known that LPS-mediated activation of TLRs subsequently activates transcription factors, nuclear factor κB (NF-κB) and activator protein 1 (AP-1), and mitogen-activated protein kinase (MAPK) signaling pathways [3]. During inflammation, cyclooxygenase-2 (COX-2) catalyzes the conversion of arachidonic acid to prostaglandins, inflammatory mediators, such as prostaglandin E_2_ (PGE_2_) [4]. Inducible nitric oxide synthase (iNOS) plays a major role in upregulating nitric oxide (NO) levels in inflammatory response [5]. Overexpressions of iNOS and COX-2 are related to the production of two major inflammatory mediators, NO and PGE_2_, and these results lead to the development of inflammatory diseases [4,5,6]. Additionally, proinflammatory cytokines, including tumor necrosis factor-a (TNF-α), interleukin-6 (IL-6), interleukin-8 (IL-8), and interleukin-1β (IL-1β), can significantly promote the progression of inflammation [7].

Therefore, in the treatment of inflammation, it is very important to select a candidate substance that inhibits inflammatory modulators. While there are several types of synthetic drugs for the treatment of inflammatory diseases, in the consideration of human health, bioactive compounds or natural product extracts still play an important role in the prevention and treatment of inflammatory diseases [8].

The genus *Bidens* (Asteraceae) is distributed worldwide and has been used historically in traditional Asian medicine. All parts of *Bidens* are utilized for the treatment of 40 disorders [9], for example as an antipyretic and analgesic and for the treatment of gastrointestinal hemorrhage and eczema [9]. Pharmacological studies have shown that *Bidens pilosa* L. extract (*B. pilosa* extract) possesses a wide range of biological activities, which include antioxidative [10], anticancer [11], antidiabetic [12], anti-inflammatory [13], antimicrobial [14], and immunomodulatory activity [15]. However, the specific active compounds responsible for the individual pharmacological effects of *B*. *pilosa* extract have not been sufficiently researched. In this study, we isolated a flavonoids-type phytochemical, isookanin, from *B. pilosa* through bioassay-guided fractionation, based on its capacity to inhibit inflammation. Isookanin (C_15_H_12_O_6_, molecular weight: 288.3) is a phenolic flavonoid presented in the *Bidens* extract [16]. It has been reported that isookanin possesses some biological properties, such as antioxidative [17,18], anti-diabetic [17], and inhibitory of α-amylase [16]. However, the anti-inflammatory mechanism of isookanin has not yet been studied. Therefore, in this study, we identified isookanin as an anti-inflammatory component in an aqueous extract of *B. pilosa* and explored the anti-inflammatory effects and the related mechanisms of isookanin in LPS-activated macrophages.

## 2. Results

### 2.1. Isolation of Anti-Inflammatory Active Phytochemical from Bidens pilosa L.

To explore natural anti-inflammatory substances, we evaluated the inhibitory effect of *B. pilosa* extracts and the fractions on inflammation, by measuring the NO and PGE_2_ production after pretreatment in LPS-stimulated RAW 264.7, mouse macrophages. NO and PGE_2_ production were measured by the Griess reagent and PGE_2_ enzyme-linked immunosorbent assay (ELISA), respectively. A selective inhibitor of inducible nitric oxide, *N*-(3-(aminomethyl)benzyl)acetamidine (1400 w), was used as a positive control for NO inhibition, and a cyclooxygenase (COX-2) selective nonsteroidal anti-inflammatory drug (NSAID), celecoxib, was used as a positive control for PGE_2_ inhibition. *B. pilosa* was extracted using water and bioassay-guided fractionation was carried out as depicted in Figure 1A. The aqueous extract of the *B. pilosa* could significantly inhibit NO and PGE_2_ production by 58% and 42%, respectively, at a concentration of 50 μg/mL (Figure 1B,C). The extracts were divided into fractions using ethyl acetate (EtOAc), *n*-butanol (*n*-BuOH), and H_2_O, depending on the solvent polarity. Among them, the EtOAc fraction showed the most potent anti-inflammatory activity with 91% NO inhibition and 82% PGE_2_ inhibition at a concentration of 50 μg/mL (Figure 1B,C). Thus, the EtOAc fraction was further separated using silica gel, Sephadex LH-20 column chromatography, and preparative HPLC to yield single compound protocatechuic acid (1) and isookanin (2) (Figure 2A–C). The structure of compound 2 was elucidated using HR-ESI MS/MS and NMR spectroscopy (Table 1 and Table 2). The fragmentation patterns of compound 2 showed molecular ions at *m*/*z* 269 [M − H − H_2_O]^−^, 151 [M − H − C_8_H_8_O_2_]^−^, 135 [M − H − C_7_H_4_O_4_]^−^, and 107 [M − H − C_8_H_8_O_2_ − CO_2_]^−^ (Appendix A). In ^1^H-and ^13^C-NMR spectral data, the flavanone skeleton was confirmed for a carbonyl group at δ_C_ 194.0, an oxygenated methine (C-2) at δ_C_ 81.6, and a methylene (C-3) at δ_C_ 45.1, with signals for the ABX-coupled system at δ_H_ (5.37, 3.06, and 2.72) (Table 2 and Appendix A). The hydroxy groups of two catechol moieties were also assigned at C-7, -8, -3′, and -4′ in A and B ring. Based on this data, compound 2 was confirmed as isookanin [16]. Isookanin (2) showed an excellent inhibitory effect of NO and PGE_2_ production compared with protocatechuic acid (1) at a concentration of 10 μg/mL (Figure 1D,E). These results might attribute isookanin to be an anti-inflammatory component in aqueous extract of *B. pilosa.*

### 2.2. Effect of Isookanin on NO and PGE_2_ Production in LPS-Induced RAW 264.7 Cells

To evaluate the inhibitory effect of isookanin on the production of proinflammatory mediators, we measured the production of NO and PGE_2_ in the LPS-stimulated RAW 264.7 cells and cell viability studies were first conducted to confirm the cytotoxic effects of isookanin administration. RAW 264.7 cells were pretreated with isookanin for 2 h, and then stimulated with LPS (100 ng/mL) for 24 h. The cell viability measured using the EZ-Cytox™ reagent and the results demonstrated that isookanin did not cause cytotoxicity to the RAW 264.7 cells at concentrations ranging from 1 to 10 µg/mL, which could be used in further studies (Figure 3A). Treatment with LPS alone significantly increased the levels of NO, whereas isookanin dose-dependently decreased the LPS-induced NO production in RAW 264.7 cells (Figure 3B) and 1400 w (500 nM) was used as a positive control as a NO inhibitor. LPS stimulation also increased PGE_2_ production, but isookanin effectively inhibited the generation of PGE_2_ induced by LPS (Figure 3C). Celecoxib (0.1 nM) was used as a positive control as a PGE_2_ inhibitor and when 10 µg/mL of isookanin was treated, the PGE_2_ inhibitory effect was similar to that of the positive control group. The NO and PGE_2_ inhibitory effects of isookanin at 10 µg/mL were 72% and 57%, respectively. These results demonstrate that isookanin exhibits an anti-inflammatory effect by inhibiting proinflammatory mediators in LPS-stimulated RAW 264.7 cells.

### 2.3. Effect of Isookanin on iNOS and COX-2 Expression in LPS-Induced RAW 264.7 Cells

iNOS is primarily responsible for producing NO in inflammatory processes and is not typically expressed in resting cells, but is induced by certain cytokines or microbial products. In addition, in the inflammatory process, the synthesis of PGE_2_ is mainly regulated by the local expression and activity of COX-2 [19]. In this study, we examined the effect of isookanin on the LPS-induced expression of iNOS and COX-2 using the luciferase reporter assay. The results revealed that LPS-only treated cells showed increased expression of iNOS and COX-2, while the cells pretreated with isookanin before LPS stimulation showed a significant and dose-dependent decrease in iNOS and COX-2 expression; the reduction effects were 51.3% and 36.5% with 10 µg/mL concentration, respectively (Figure 4A,B). The inhibitory effects of isookanin on NO and PGE_2_ are closely related to iNOS and COX-2 expressions, respectively, and these results indicate that isookanin regulates the production of NO and PGE_2_ by inhibiting the transcriptional activity of iNOS and COX-2.

### 2.4. Effects of Isookanin on LPS-Induced NF-κB and AP-1 Activation in RAW 264.7 Cells

The activation of transcription factors, nuclear factor κB (NF-κB) and activator protein 1 (AP-1), regulates the expression of the target genes that are involved in inflammation. These two transcription factors play important roles in the gene expression of iNOS and COX-2. We investigated if isookanin exerts anti-inflammatory activities by affecting these two pathways, and we examined the effect of isookanin on LPS-induced AP-1 and NF-κB activity using the luciferase reporter assay. Figure 5 shows that isookanin significantly suppressed the AP-1 signaling pathway (Figure 5A), but had little effect on the NF-κB signaling pathway (Figure 5B).

### 2.5. Effect of Isookanin on Phophorylation of MAPKs in LPS-Induced RAW 264.7 Cells

Next, we investigated the MAPKs, which act upstream of AP-1, to explore the signaling pathways that regulate the inflammation of isookanin. The MAPK signaling pathway is essential to regulating many cellular processes, including inflammation. The regulatory role of MAPKs in the synthesis of inflammatory mediators makes it a potential target of anti-inflammatory therapeutic agents [20]. To investigate whether these signaling pathways are involved in the isookanin effects, we performed Western blot analysis for MAPKs (Extracellular Signal-Regulated Kinases 1/2 (ERK1/2), c-jun NH2-terminal kinase (JNK), and p38 MAPK). Figure 6 shows that the phosphorylation of MAPKs was induced by LPS (200 ng/mL). In addition, the phosphorylation of p38 MAPK and JNK was inhibited by isookanin, whereas isookanin had slight effect on that of ERK1/2. These results suggest that isookanin particularly attenuates the phosphorylation of p38 MAPK and JNK in the LPS-induced activation of MAPKs. In addition, this may indicate that isookanin inhibits the production of inflammatory mediators by inhibiting the activation of transcription factors AP-1 through the MAPK signaling pathways.

### 2.6. Effect of Isookanin on LPS-Induced Proinflammatory Cytokines Production in THP-1 Cells

To further investigate the anti-inflammatory effect of isookanin, the levels of TNF-α, IL-6, IL-8, and IL-1β in the culture supernatant were determined by ELISA. Figure 7 shows that LPS stimulation significantly upregulated the production of the proinflammatory cytokines in THP-1, human macrophages. However, treatment with isookanin dose-dependently downregulated the levels of TNF-α, IL-6, IL-8, and IL-1β induced by LPS. These data suggest that isookanin exerts anti-inflammatory activity via suppression of proinflammatory cytokines in LPS-stimulated THP-1 cells.

## 3. Discussion

Inflammation is a defense response to stimuli and manifests as redness, swelling, heat, pain, and dysfunction. Excessive inflammation can lead to diseases of such kinds as arthritis [21], colitis [22], and asthma [23]. After stimulation, monocytes can differentiate macrophages at the site of infection and release inflammatory factors, such as nitric oxide (NO), prostaglandin E_2_ (PGE_2_), and other cytokines [24]. Therefore, what is vital in treating inflammatory diseases is to control inflammatory factors. In recent years, plant therapy has been highly valued, because bioactive compounds or extracts derived from natural products still play an important role in human health in the prevention and treatment of inflammatory disorders [8]. *B. pilosa* extract was known to have anti-inflammatory effects [13], but the specific active compounds responsible for the individual pharmacological effects of *B. pilosa* extract have not yet been fully uncovered. Although there are existing papers that have analyzed components [25,26,27], there were few papers on the separation and analysis of components according to their physiological activity, so their physiological activities are often unclear. In this study, the active ingredients from the aqueous extract of *B. pilosa* were isolated through a bioassay-guided fractionation, based on its ability to inhibit inflammation. The active fraction was finally determined through the bioactivity screening process, and the components contained in *B. pilosa* extract were analyzed and identified. Six compounds, namely, protocatechuic acid (1), isookanin (2), (−)-quinic acid (3), 3-*O*-caffeoylquinic acid (4), 4-*O*-caffeoylquinic acid (5), and 5-*O*-caffeoylquinic acid (6) were separated in the aqueous extract through the extraction process and identified by HR-ESI MS/MS analysis (Table 1 and Appendix A) [28,29]. Among these ingredients, isookanin was confirmed to be effective in anti-inflammatory activity and was finally determined as an active ingredient. Although isookanin is a component that is known as a phenolic flavonoid component contained in some plants, such as *Asteraceae* [16] and *Leguminosae* [18], its anti-inflammatory efficacy and mechanism of action have not previously been accurately identified. Therefore, this study revealed the anti-inflammatory effect of isookanin isolated from *B. pilosa*.

The structure of isookanin showed that the hydroxy groups of two catechol moieties were also assigned at C-7, -8, -3′, and -4′ in A and B rings. Many of its structurally similar analogues, such as apigenin, acacetin, chrysin, baicalein, wogonin, luteolin, and velutin, were reported to suppress proinflammatory mediators in LPS-stimulated macrophages [30,31,32,33,34,35,36]. Analysis of the structure–activity relationships of flavones showed that the anti-inflammatory effects could be enhanced by the 5- or 7-hydroxyl groups on the A ring, or 3′-hydroxyl groups on the B ring [37]. The chemical structure of isookanin includes these dispositions, such as the 7-OH and 3′-OH. Indeed, our results show that isookanin potently decreases LPS-induced proinflammatory mediators.

LPS-stimulated macrophages release various proinflammatory mediators, including NO and PGE_2_, as well as proinflammatory cytokines, such as TNF-α, IL-1β, IL-6, and IL-8 [38]. All these substances promote the progression of inflammation and exacerbate inflammation through synergistic interplay with other inflammatory mediators [39], and this may cause pathological changes and tissue damage [38]. Elevated expression of NO and PGE_2_ is often associated with a variety of inflammatory diseases, such as asthma [24] and colitis [40]. Proinflammatory cytokines have been described as upregulation linked to the pathogenesis of numerous inflammatory diseases, including cancer [41]. Our study demonstrated that isookanin effectively suppressed the production of NO and PGE_2_ in LPS-stimulated RAW 264.7 cells, at non-toxic concentrations. Furthermore, isookanin significantly suppressed the LPS-induced expression of iNOS and COX-2 that are key enzymes in the synthesis of NO and PGE_2_, respectively. It is known that LPS-induced proinflammatory mediators are mainly regulated by NF-κB and AP-1 [42,43]; our results indicate that isookanin significantly inhibits the AP-1 signaling pathway (Figure 5A), but had little effect on the NF-κB signaling pathway (Figure 5B).

Since MAPK signaling mediates the expression of inflammatory mediators through the activation of transcription factors, such as AP-1 and NF-κB, it makes them potential targets for anti-inflammatory therapeutics [21]. Therefore, the effect of isookanin on LPS-induced phosphorylation of p38 MAPK, ERK1/2, and JNK in RAW 264.7 cells was assessed. The results indicate that isookanin suppresses the phosphorylation of p38 MAPK and JNK in a concentration-dependent manner, but has little effect on ERK1/2 phosphorylation (Figure 6). According to previous studies, p38 MAPK regulates transcriptional and post-transcriptional levels of iNOS and COX-2 [44,45]. Thus, isookanin may act not only at transcription but also at post-transcriptional levels, such as affecting mRNA stability. However, this study did not confirm the mRNA stability of iNOS and COX-2 at the post-transcriptional level through inhibition of p38 MAPK activation. Therefore, whether p38 MAPK affected by isookanin directly affects the post-transcriptional levels of iNOS and COX-2 needs to be confirmed by further works for accurate understanding.

We investigated the molecular mechanism of isookanin in the inflammatory response, and the results show that isookanin reduced AP-1 and phospho-p38 MAPK and phospho-JNK activation. These results are similar to those previously published by Gao et al. [46] in a study on the anti-inflammatory efficacy of genkwanin: pretreatment with genkwanin significantly inhibited AP-1 expression, but had little effect on NF-κB expression. In addition, that previous study showed genkwanin reduced LPS-induced phospho-p38 MAPK and phospho-JNK levels in MAPK’s signaling pathway, but phospho-ERK1/2 confirmed that it had no effect. As mentioned in a previously published paper, these results are likely to be associated with the increased expression of the mitogen-activated protein kinase phosphatase-1 (MKP-1), a stress-responsive protein. MKP-1 protein is located in the nucleus through its N terminus, and preferentially dephosphorylates activated p38 MAPK and JNK, compared to ERK1/2 [46,47]. It is a key negative regulator of macrophages in response to inflammatory stimuli and is responsible for blocking the production of proinflammatory cytokines [48,49]. Therefore, the differential regulations of isookanin on MAPK phosphorylation suggest that the upstream regulator may be MKP-1. We plan to explore mechanism studies on the correlation with MKP-1 in future studies.

## 4. Materials and Methods

### 4.1. Plant Materials

The aerial part of *Bidens pilosa* L. was collected from Hongcheon-gun, Gangwon-do, Korea in November 2018. The plant was authenticated by Ph.D. W.K. Whang of the Pharmacal Resources Botany Laboratory, College of Pharmacy, Chung-Ang University, where a voucher specimen (CAU-PBL-20180930) was deposited.

### 4.2. Materials

Dulbecco’s Modified Eagle Medium (DMEM) and fetal bovine serum were purchased from Gibco (Grand Island, NY, USA). RPMI 1640 medium came from Cytiva (Marlborough, MA, USA), and penicillin/streptomycin antibiotics came from Invitrogen (Carlsbad, CA, USA). EZ-Cytox reagent was obtained from DoGenBio (Seoul, South Korea). Griess reagent, dimethyl sulfoxide (DMSO), protocatechuic acid, and lipopolysaccharide (LPS) were purchased from Sigma-Aldrich (St. Louis, MO, USA). The ELISA kits for PGE_2_, TNF-α, IL-6, IL-8, and IL-1β were purchased from R&D Systems, Inc. (Minneapolis, MN, USA). Isookanin was obtained from ChemFaces Biochemical Co. Ltd. (Wuhan, Hubei, China), and dissolved in DMSO. The luciferase assay system and lipofectamine™ 3000 reagent were purchased from Promega Co. (Madison, TN, USA) and Invitrogen (Waltham, MA, USA), respectively. Mitogen-activated protein kinases (MAPKs) antibodies (*p*-ERK1/2, ERK1/2, *p*-p38, p38, *p*-SAPK/JNK, SAPK/JNK) were purchased from Cell Signaling Technology, Inc. (Danvers, MA, USA), while secondary antibodies were purchased from Santa Cruz Biotechnology, Inc. (Dallas, TX, USA).

### 4.3. Extraction and Bioactivity-Guided Isolation

The dried aerial parts of *B. pilosa* (4 kg) were extracted by boiling in water (3 × 100 L) for 3 h each, followed by vacuum concentration. This crude extract (390.8 g) was suspended in H_2_O and successively partitioned with EtOAc (ethyl acetate), *n*-BuOH (*n*-buthanol), and H_2_O to obtain the EtOAc (16.02 g), *n*-BuOH (32.75 g), and H_2_O (122.32 g) fractions, after removal of the solvents in vacuo. The ability of the fractions to inhibit NO and PGE_2_ production in LPS-treated RAW 264.7 cells was evaluated. The most active EtOAc fraction (16.02 g) was subjected to silica gel column chromatography eluted with CHCl_3_–CH_3_OH (20:1—5:1, 0:1, *v*/*v*) to afford five fractions (CC1–CC5). Fraction CC1 was separated by Sephadex LH-20 column chromatography with CHCl_3_–MeOH (1:1) to yield sub-fractions CC11 and CC12. Sub-fraction CC12 was applied to semi-preparative HPLC (20% CH_3_CN in 0.1% TFA, 10 mL/min) to yield compounds 1 (8.7 mg, tR = 8.5 min) and 2 (10.3 mg, tR = 17.6 min), respectively.

### 4.4. Chemical Analysis

The NMR spectra were recorded on a Bruker Avance III 500 instrument. HR-ESI MS/MS was acquired in positive and negative ion modes on an AB SCIEX QTOF 5600 instrument. The HPLC analysis was performed on a Waters 2695 Separation Module with a Waters 2996 Photodiode Array Detector (Waters Corporation, Milford, CT, USA), using a Phenomenex Luna C_18_ column (4.6 × 250 mm, 5 μm). The preparative HPLC equipment was a Waters Prep LC 2000 equipped with a Waters 2487 Dual λ Absorbance detector (Waters Corporation, Milford, CT, USA). The column applied in this work was a Phenomenex Luna C_18_ column (21.1 × 250 mm, 5 μm). Column chromatography was conducted using silica gel 60 (Merck, Darmstadt, Germany; 230–400 mesh) and Sephadex LH-20 (GE Healthcare Life Sciences (Little Chalfont, BU, UK). Thin-Layer Chromatography (TLC) was performed with silica gel HLF 250 μm, and the spots were detected by spraying the plates with 10% H_2_SO_4_–EtOH reagent followed by heating at 120 °C for 5 min.

### 4.5. Cell Culture

The mouse macrophage cells (RAW 264.7) were obtained from the Korean Cell Line Bank (Seoul, South Korea) and maintained in Dulbecco’s Modified Eagle’s Medium (DMEM), while human leukemia monocytic cells (THP-1) were obtained from the Korean Cell Line Bank (Seoul, South Korea) and maintained in RPMI 1640 medium, containing 10% fetal bovine serum (FBS) and 1% penicillin/streptomycin at 37 °C in a humidified 5% CO_2_ atmosphere.

### 4.6. Cell Viability Assay

The cell viability assay was carried out using EZ-Cytox reagent. RAW 264.7 cells were plated at a density of 2 × 10^5^ cells/well in 24-well plates and incubated for 24 h. The cells were subsequently treated with various concentrations of the samples for 24 h. At the end of the treatment period, 10 µL of the cell viability assay solution was added to each well, and the plate was incubated for 2 h. The absorbance at 450 nm was measured using a microplate reader (BioTek Instruments, Inc., Winooski, VT, USA). The results are expressed as a percentage of the control.

### 4.7. Measurement of Nitric Oxide Production

RAW 264.7 cells were seeded at 2 × 10^5^ cells per well in 24-well culture plates. The cells were incubated in the presence or absence of samples for 2 h and then treated with LPS (100 ng/mL) for 24 h. Cell culture supernatant (50 µL) was incubated with equal volume of Griess reagent (equal volumes of 1% (*w*/*v*) sulfanilamide in 5% (*v*/*v*) phosphoric acid and 0.1% (*w*/*v*) naphtylethylene) for 10 min at room temperature. The absorbance was measured by spectrometry at 540 nm wavelength and calculated against a sodium nitrite standard curve [50].

### 4.8. Measurement of PGE2, TNF-α, IL-6, IL-8, and IL-1β by Enzyme-Linked Immunosorbent Assay (ELISA)

The levels of PGE_2_ in the RAW 264.7 macrophages and the cytokines (TNF-α, IL-6, IL-8, and IL-1β) in the THP-1 cells were measured using a commercial ELISA kit according to the manufacturer’s instructions. RAW 264.7 cells were seeded at 2 × 10^5^ cells per well in 24-well culture plates. The cells were treated with the samples and then induced by LPS (100 ng/mL) for 24 h. The cell culture supernatants were collected and assayed for PGE_2_. THP-1 cells were primed for differentiation with 200 nM PMA and seeded in a 24-well plate (2.5 × 10^5^ cells/well) and cultured overnight. They were then treated first with isookanin (1, 5, and 10 µg/mL) and induced by LPS (5 ng/mL) for 24 h. The supernatant was harvested and analyzed to measure the levels of TNF-α, IL-6, IL-8, and IL-1β, respectively. The PGE_2_, TNF-α, IL-6, IL-8, and IL-1β concentrations were determined using a standard curve. All samples and standards were measured in duplicate.

### 4.9. Plasmids

The mouse COX-2 promoter region containing 963 bp (−963/+1) was cloned into pGL3basic vector (Promega, Madison, TN, USA). The mouse iNOS promoter region containing 1742 bp (−1741/+1) was cloned into pGL3basic vector (Promega). The reporter plasmids, pNF-κB-Luc and pAP-1-Luc (PathDetect Cis-Reporting System), were purchased from Agilent Technologies (La Jolla, CA, USA).

### 4.10. Transient Transfection and Luciferase Assay

The RAW 264.7 cells were transfected with the mouse iNOS, mouse COX-2, AP-1, and NF-κB luciferase reporters using lipofectamine™ 3000 reagent (Invitrogen, Waltham, MA, USA), according to the manufacturer’s instructions. A plasmid containing the Renilla luciferase gene without promoter (pRL Renilla Luciferase Control Reporter Vector, Promega) was co-transfected with the firefly luciferase reporter gene and used as a normalization control. After 2 h of transfection, the cells were incubated in the presence or absence of isookanin (1, 5, and 10 μg/mL) and induced by LPS (200 ng/mL) for 24 h. The cells were then harvested and lysed by luciferase cell culture lysis reagent, and the supernatant was assayed for their luciferase activity using a dual luciferase assay system (Promega) and Infinite 200 PRO luminometry (Tecan, Männedorf, Switzerland). Renilla luciferase expression was normalized to target gene expression.

### 4.11. Western Blotting

Western blotting was performed to measure the levels of proteins associated with the MAPKs pathways. RAW 264.7 cells were pretreated with isookanin (1, 5, and 10 μg/mL) for 2 h, and induced by LPS (200 ng/mL) for 20 min. The cells were then harvested, and lysed by protein extraction solution, PRO-PREP™ (iNtRON Biotechnology, Seongnam, South Korea). The protein extracts (20 μg) from lysed cells were loaded on a NuPAGE Novex 10% Bis-Tris Gel 1.0 mm, 15-well (Invitrogen, Waltham, MA, USA) in each lane, and transferred to a nitrocellulose membrane by using iBlot™ Gel Transfer Device (Invitrogen, Waltham, MA, USA). The membranes were blocked with 5% bovine serum albumin (BSA) for 1 h and then incubated with primary antibodies (1:1000 dilution of stock) in 10 mL of blocking buffer, followed by incubation with horseradish peroxidase-conjugated anti-mouse or rabbit IgG secondary antibody (1:2000 dilution of stock). The protein bands were detected by enhanced chemiluminescence detection reagents (Invitrogen, Waltham, MA, USA) and visualized using ImageQuant™ LAS 500 (Cytiva, Marlborough, MA, USA).

### 4.12. Statistical Analysis

All experimental data are expressed as mean ± standard deviation. Differences between the control and the treatment group were evaluated by one-way ANOVA (SPSS, IBM, Armonk, NY, USA). *p* < 0.05, 0.01, or 0.001 were considered statistically significant.

## 5. Conclusions

In conclusion, isookanin was isolated from the aqueous extract of *B. pilosa* via bioassay-guided fractionation as an active compound for anti-inflammation. Our results show that isookanin downregulates NO and PGE_2_ production in LPS-exposed RAW 264.7 cells. The inhibitory activity of isookanin occurred by the suppression of iNOS and COX-2 expression via the reduction of AP-1 activation. This anti-inflammatory effect was linked to a MAPK signaling pathway, and isookanin inhibited phosphorylation of p38 MAPK and JNK (Figure 8). In addition, we also confirmed that isookanin downregulated the production of TNF-α, IL-1β, IL-6, and IL-8 induced by LPS in THP-1 cells. Based on these results, isookanin shows the prospect of being a therapeutic agent for inflammatory diseases, and the therapeutic effect of isookanin as an anti-inflammatory agent needs to be confirmed in vivo using established animal models of inflammation.

## Figures and Tables

**Figure 1 molecules-26-00255-f001:**
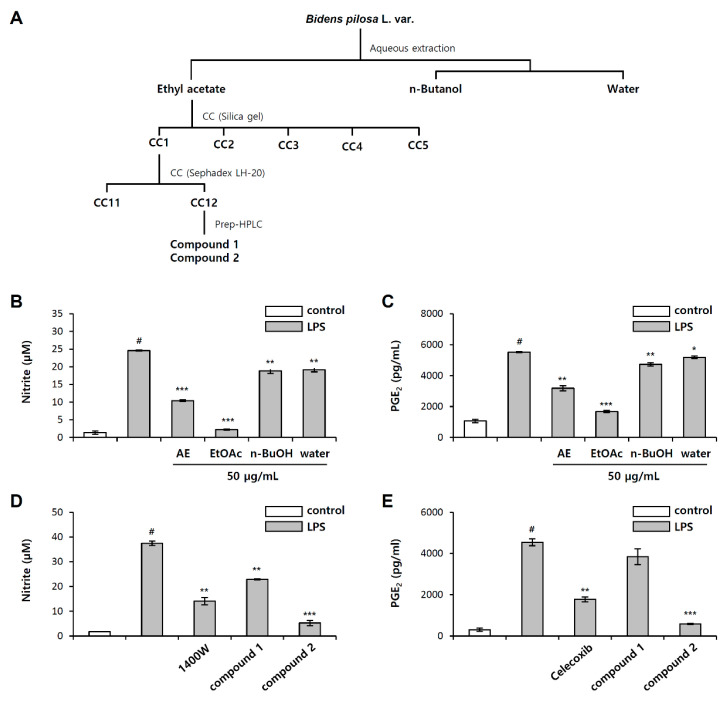
(**A**) Bioassay-guided isolation scheme of isookanin from *B. pilosa*. Effect of the *B. pilosa* aqueous extract fractions on (**B**) nitric oxide (NO), and (**C**) prostaglandin E2 (PGE_2_) production in lipopolysaccharide (LPS)-induced RAW 267.4 cells. Effect of compounds 1 and 2 on (**D**) NO, and (**E**) PGE_2_ production in LPS-induced RAW 267.4 cells. The results are mean ± standard deviation (SD) (*n* = 3). # *p* < 0.01 vs. LPS-untreated control. * *p* < 0.05, ** *p* < 0.01, and *** *p* < 0.001 vs. LPS-treated control. AE: aqueous extract; EtOAc: ethyl acetate partition; n-BuOH: *n*-butanol partition; Water: water partition.

**Figure 2 molecules-26-00255-f002:**
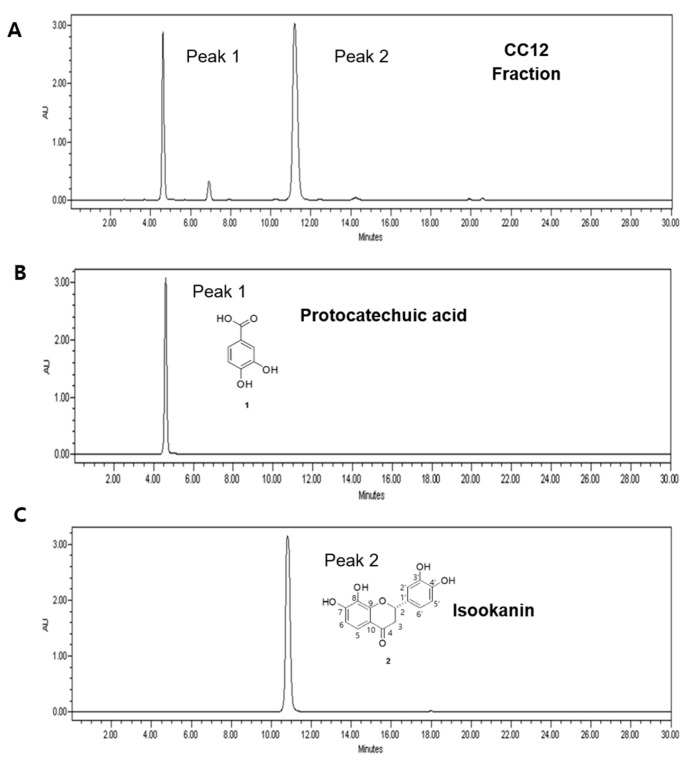
(**A**) HPLC profiles of sub-fraction (CC12) separated by column chromatography. Use of preparative HPLC to yield (**B**) single compound 1 protocatechuic acid and (**C**) compound 2 isookanin.

**Figure 3 molecules-26-00255-f003:**
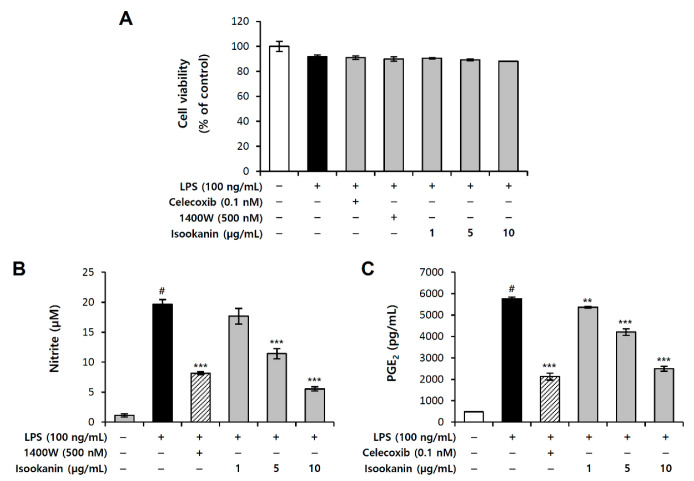
Inhibitory effect of isookanin on NO and PGE_2_ production in LPS-induced RAW 267.4 cells. Cells were pretreated with the indicated concentrations of isookanin or positive control for 2 h, before incubation with LPS (100 ng/mL) for 24 h. (**A**) Cell viability was measured using the WST-1 assay. (**B**) The levels of NO in the culture medium were measured using the Griess reaction. (**C**) The levels of PGE_2_ in the culture medium were measured using an enzyme-linked immunosorbent assay (ELISA) kit. The results are mean ± standard deviation (SD) (*n* = 3). # *p* < 0.01 vs. LPS-untreated control. ** *p* < 0.01, and *** *p* < 0.001 vs. LPS-treated control.

**Figure 4 molecules-26-00255-f004:**
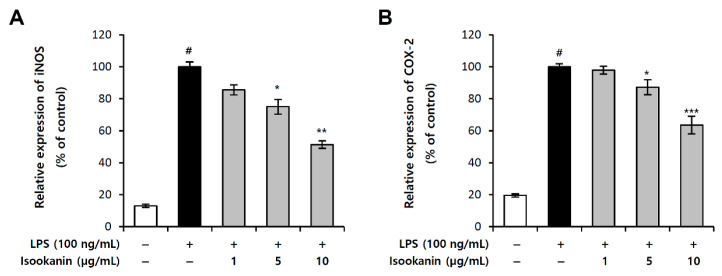
Effects of isookanin on inducible nitric oxide synthase (iNOS) and cyclooxygenase-2 (COX-2) expression in LPS-induced RAW 264.7 cells. The (**A**) iNOS, and (**B**) COX-2 luciferase reporter vector was transfected into RAW 264.7 cells, and cultured for 24 h. The cells were pretreated with isookanin for 2 h, and then stimulated with LPS (200 ng/mL). Luciferase activity was calculated against an LPS-unstimulated control. The results are mean ± standard deviation (SD) (*n* = 3). # *p* < 0.01 vs. LPS-untreated control. * *p* < 0.05, ** *p* < 0.01, and *** *p*< 0.001 vs. LPS-treated control.

**Figure 5 molecules-26-00255-f005:**
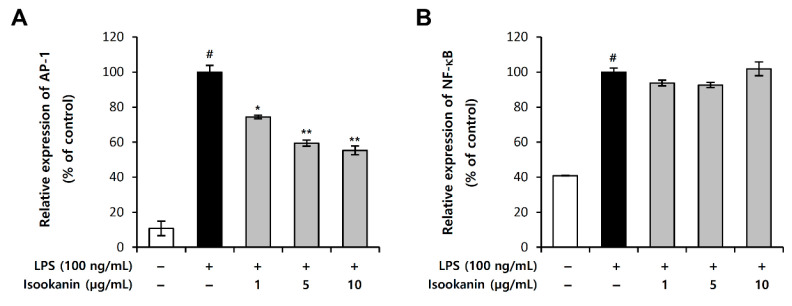
Effects of isookanin on activator protein 1 (AP-1) and nuclear factor κB (NF-κB) expression in LPS-induced RAW 264.7 cells. The (**A**) AP-1 and (**B**) NF-κB luciferase reporter vector was transfected into RAW 264.7 cells and cultured for 24 h. The cells were pretreated with isookanin for 2 h and then stimulated with LPS (200 ng/mL). Luciferase activity was calculated against an LPS-unstimulated control. The results are mean ± standard deviation (SD) (*n* = 3). # *p* < 0.01 vs. LPS-untreated control. * *p* < 0.05, and ** *p* < 0.01 vs. LPS-treated control.

**Figure 6 molecules-26-00255-f006:**
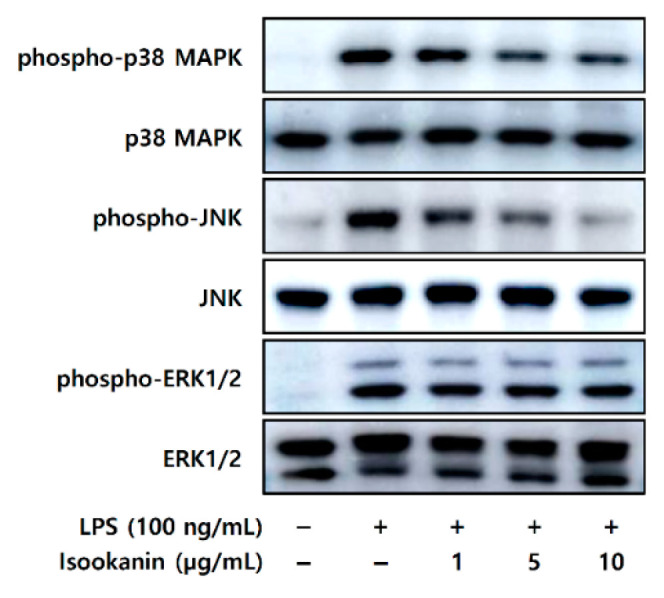
Effects of isookanin on the phosphorylation of mitogen-activated protein kinases (MAPKs) in LPS-induced RAW 264.7 cells. The RAW 264.7 cells were cultured for 24 h, and then treated with LPS (200 ng/mL) in the presence or absence of isookanin. Cell lysates were prepared at 20 min, and then subjected to Western blot analysis. The bands for phospho-ERK1/2, phospho-p38 MAPK, and phospho-JNK were detected and normalized to their total forms of MAPKs.

**Figure 7 molecules-26-00255-f007:**
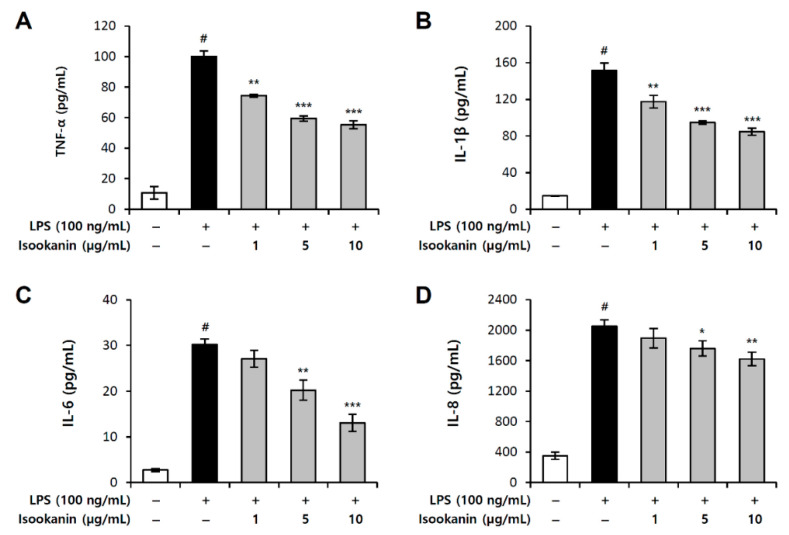
Inhibitory effect of isookanin on LPS-induced tumor necrosis factor-a (TNF-α), interleukin-6 (IL-6), interleukin-8 (IL-8), and interleukin-1β (IL-1β) production in THP-1 cells. Cells were pretreated with the indicated concentrations of isookanin for 2 h, before incubation with LPS (5 ng/mL) for 24 h. The levels of (**A**) TNF-α, (**B**) IL-1β, (**C**) IL-6, and (**D**) IL-8 in the culture medium were measured by an ELISA kit. The results are mean ± standard deviation (SD) (*n* = 3). # *p* < 0.01 vs. LPS-untreated control. * *p* < 0.05, ** *p* < 0.01, and *** *p* < 0.001 vs. LPS-treated control.

**Figure 8 molecules-26-00255-f008:**
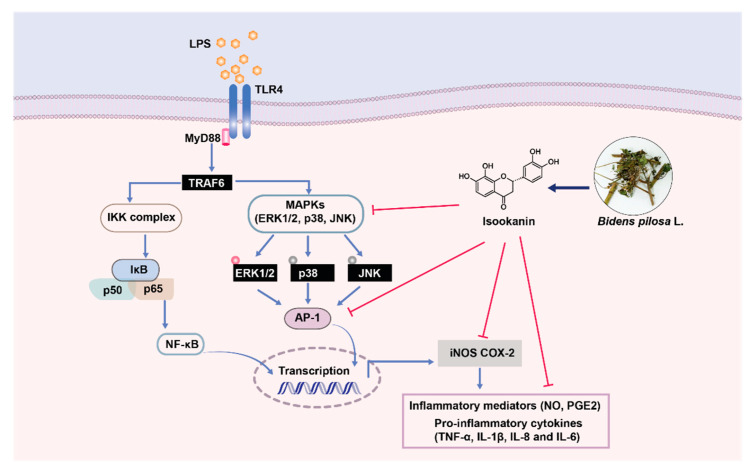
Anti-inflammatory mechanism of isookanin isolated from *Bidens pilosa* L. is associated with suppressing the TLR4-mediated AP-1 and MAPK signaling pathways induced by LPS.

**Table 1 molecules-26-00255-t001:** HR-ESI MS/MS data for compounds isolated in the aqueous extract of *B. pilosa*. Triple TOF 5600+ (AB SCIEX, Washington, DC, USA), range (100–2000) *m*/*z*, ion spray voltage 4500 V, collision energy 35 eV.

Compounds		^a^ [M + H]^+^/^b^ [M − H]^−^ (*m*/*z*)		MS^2^ Fragments (*m*/*z*)
		Observed *m*/*z*	Calculated *m*/*z*	
1	Protocatechuic acid	155.0340 ^a^	155.0344 ^a^	137, 109, 93, 81, 65
2	Isookanin	287.0568 ^b^	287.0556 ^b^	269, 151, 135, 107
3	(−)-quinic acid	191.0565 ^b^	191.0556 ^b^	127, 109, 93, 85
4	3-*O*-caffeoylquinic acid	353.0876 ^b^	353.0873 ^b^	191, 179, 135
5	4-*O*-caffeoylquinic acid	353.0884 ^b^	353.0873 ^b^	191, 179, 173, 135
6	5-*O*-caffeoylquinic acid	353.0886 ^b^	353.0873 ^b^	191

^a^: Detection in the positive mode; ^b^: Detection in the negative mode.

**Table 2 molecules-26-00255-t002:** ^1^H and ^13^C NMR data (*δ* in ppm, *J* in Hz) of isookanin in methanol-d_4_ (500 MHz).

Position	Compound 2	
	*δ* _C_	*δ*_H_ (*J* in Hz)
2	81.6	5.37, dd (12.4, 2.9)
3	45.1	ax 3.06, dd (16.9, 12.4), eq 2.72 (16.9, 3.0)
4	194.0	
5	119.3	7.30, d (8.7)
6	110.9	6.52, d (8.7)
7	154.0	
8	134.0	
9	152.7	
10	115.7	
1′	131.9	
2′	115.0	6.98, d (1.9)
3′	146.5	
4′	146.9	
5′	116.2	6.78, d (8.1)
6′	119.6	6.85, dd (8.1, 1.9)

## Data Availability

The data presented in this study are available on request from the corresponding author.

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
