# Peer review of "Anti-Inflammatory Activity and Mechanism of Isookanin, Isolated by Bioassay-Guided Fractionation from Bidens pilosa L."

_molecules, 2021, doi:10.3390/molecules26020255_

Round 1

Reviewer 1 Report

Xin et al. describe the anti-inflammatory activity of the natural compound isookanin, which reduces nitric oxide (NO) and PGE2 formation in murine RAW 264.7 macrophages and the expression of a number of pro-inflammatory cytokines in  human THP-1 cells. It is a nice and comprehensive study but several issues have to be addressed before publication. 

A)

In figure 3A a slight decrease in cell viability occurs at 10 µg/ml Isookanin. This should be stated in the text. Please add statistical analyses to this figure.

B)

Absolut no information about the reporter gene constructs used in their transfection experiments are presented. Are these murine or human promoter fragments?   Are these full-length promoter constructs or shortened fragments of the promoters?

Which construct was used to normalize for the transfection efficacy? A renilla reporter plasmid or something else?

C)

It is well known that iNOS and COX2  are regulated by transcriptional and post-transcriptional mechanisms. Post-transcriptional regulation of iNOS and Cox2 mRNA stability is mediated, at least in part, via the p38-MAPK pathway. Therefore, the possibility exist, that isookanin also modulates post-transcriptional processes (e.g. mRNA stability) resulting in decreased NO and PGE2 expression. The authors have to discuss this point.

D)

Please add in the result section the information that RAW264.7 macrophages are murine and THP-1 are human cells.

E)

In line 162 and in the figure legend of figure 5, figure 5A is described as “relative expression of AP-1”, but figure 5A displays NF-κB activity at the moment.

F)

Which solvent was used to dissolve isookanin? Was this solvent used also as control in their experiments?

G)

Figure 6.

In the text and in the figure legend Figure 6A,B  and C are mentioned, but at the moment only one figure 6 without subfigures exist. Does the authors have an explanation why Isookanin 1 µg/ml and 5 µg/ml seem to enhance p-ERK1/2 expression? Moreover, the quality of the p-JNK blot is not satisfactory. In addition, an information is lacking whether the blots are representative for others done by the researchers or not.

H)

In the conclusion in line 372: There seems to be a word missing in the sentence.

Line 380: The authors should replace the statement: “needs to be confirmed in clinical trials” in: “needs to be confirmed in vivo using established animal models of inflammation”.

I)

The authors should add the name of the software used for their statistical analyses.

Reviewer 2 Report

Xin and collaborators investigated the anti-inflammatory effects of Bidens pilosa L. in macrophages. They identified two compounds which were suggested as responsible for the observed anti-inflammatory effects of the plant. The study was well designed and provided novel information to the research field. However, some major gaps need to be filled in order to improve the manuscript and make it suitable for publication. Major comments are as follows:

1) All experiments except for cytokine measurements were performed in RAW 264.7 macrophages. What was the rational for moving from RAW 264.7 to THP-1 cells?

2) Please provide full blots and antibody negative controls for all WB experiments. 

Round 2

Reviewer 1 Report

The authors adequately answered my questions.  Only minor modifications are necessary before publication of the manuscript.

Line 253: figure 6 instead of figure 6A

Line 343: Please add the concentration of PMA used for THP1 differentiation

Author Response

We greatly appreciate for your constructive suggestions. They have helped us improve the quality of the manuscript. The comments are listed below with our responses.

Line 253: figure 6 instead of figure 6A

Line 253: subfigures A in Figure 6 have been deleted.

Line 343: Please add the concentration of PMA used for THP1 differentiation

Line 343: we added the concentration of PMA (200 nM) used for THP1 differentiation

Reviewer 2 Report

The authors have responded all points raised. It is suggested they include the full blots as supplements of the manuscripts.  

Author Response

Reviewer 2

We greatly appreciate for your constructive suggestion. It have helped us improve the quality of the manuscript. The comments are listed below with our responses.

It is suggested they include the full blots as supplements of the manuscripts. 

The image of full length blots were included in supplementary file.